# The Influence of Intensive Nutritional Education on the Iron Status in Infants—Randomised Controlled Study

**DOI:** 10.3390/nu17193103

**Published:** 2025-09-29

**Authors:** Kinga Ilnicka-Borowczyk, Dagmara Woźniak, Małgorzata Dobrzyńska, Tomasz Podgórski, Karol Szymanowski, Anna Blask-Osipa, Klaudia Mieloszyk, Sławomira Drzymała-Czyż

**Affiliations:** 1Department of Bromatology, Poznan University of Medical Sciences, Rokietnicka 3 Street, 60-806 Poznan, Poland; kingailnickaborowczyk@gmail.com (K.I.-B.); dagmara.wozniak94@gmail.com (D.W.); mdobrzynska@ump.edu.pl (M.D.); 2Complex of Healthcare Institutions, Kosciuszki 96 Street, 64-700 Czarnkow, Poland; annablask@wp.pl; 3Department of Biochemistry, Poznan University of Physical Education, 61-871 Poznan, Poland; podgorski@awf.poznan.pl; 4Specialized Medical Care Unit for Mothers and Children, Wrzoska 1 Street, 60-663 Poznan, Poland; karol.szymanow@gmail.com; 5Central Laboratory, Karol Jonscher Clinical Hospital of the Poznan University of Medical Science, Szpitalna 27/33, 60-572 Poznan, Poland; klaudiamieloszyk@wp.pl

**Keywords:** nutritional programming, transferrin, ferritin, hepcidin, child nutrition, development

## Abstract

Background: Iron is an essential nutrient for the proper development of infants. Iron deficiency, a common cause of anemia—affects nearly half children under four years of age in developing countries. The aim of the study was to assess the impact of an intensive nutritional education program on the iron status of infants. Material and methods: The parents of 115 infants were randomly assigned to two groups: the study group, which received intensive nutritional education up to 12 months of age, and the control group, which received basic infant nutrition guidelines. Serum concentrations of iron metabolism parameters—among others hemoglobin, iron, ferritin, ferroportin, and total iron-binding capacity (TIBC)—were assessed at both the beginning and end of the study. Additionally, at the final time point, dietary intake of iron and components influencing its absorption (e.g., vitamin C, fiber, etc.) was evaluated based on food diaries completed by the parents. Results: At the end of the study, the study group showed a significantly higher level of hemoglobin (*p* = 0.0499), ferritin (*p* = 0.0067) and lower levels of TIBC (*p* = 0.0478) and ferroportin (*p* = 0.0410) compared to the control group. Moreover, infants in the study group demonstrated significantly higher intake of both iron (*p* = 0.0252) and vitamin C (*p* = 0.0458). Conclusions: Parental nutritional education contributes to improvements in iron metabolism indicators in infants.

## 1. Introduction

Iron-deficiency anaemia (IDA) affects an estimated 51% of children under the age of four in developing countries and approximately 12% in developed countries [1]. Iron is a vital nutrient that supports proper development across all stages of childhood [2]. It plays an essential role not only in the synthesis of red blood and muscle cells, but also in DNA replication and the development of the brain, nervous system, and immune system [2,3,4]. In infancy, iron deficiency has been linked to impaired memory and attention, increased risk of attention-deficit hyperactivity disorder (ADHD), dysfunctions in visual and auditory processing, and disturbances in social and emotional behaviours [2,5,6,7,8,9,10].

According to the Global Burden of Disease Study (2019), anemia imposed a substantial global burden on children under five, with a reported years lived with disability of 1253 (95% UI: 831–1831) per 100,000 population [11].

In Europe, the prevalence of iron deficiency among infants aged 6 to 12 months is estimated to range from 4% to 18% [12]. Among toddlers over 12 months of age living in developed countries (such as the United States, New Zealand, Finland, and Greece), the prevalence of iron deficiency without anemia may reach up to 30%) [13,14,15,16]. Globally, the issue of iron deficiency and the resulting anemia is even more widespread. According to WHO data, the global prevalence of IDA among preschool-aged children (0–4.99 years) is 42.6% (95% CI: 37.7–47.4). The highest prevalence was recorded in Africa, at 62.3% (95% CI: 59.6–64.8). Among all African countries, anemia was most prevalent in Burkina Faso, where the rate reached 86% (95% CI: 82–89). High rates of anemia were also observed in South-East Asia, where the prevalence of IDA was 53.8% (95% CI: 39.9–63.9), including as much as 61% (95% CI: 51–69) in Pakistan [17,18]. Both iron deficiency and its consequence—iron-deficiency anaemia—can negatively affect growth, energy levels, and motor and cognitive development [2,19]. Although iron supplementation is an effective treatment for anaemia, its use for prevention is generally discouraged due to potential adverse effects such as nausea, vomiting, constipation, dyspepsia, diarrhoea, dark stools, and dental staining [2,20,21]. Therefore, ensuring a proper diet and providing young children with appropriate amounts of iron through food appears to be of utmost importance.

According to the theory of nutritional programming, both nutrient excess and deficiency during the first 1000 days of life may have long-term consequences by influencing tissue development and reprogramming metabolic pathways [22,23,24]. Despite widespread access to nutritional guidance for children, parents continue to make significant dietary mistakes. The study by Woźniak et al. [25] revealed excessive intake of protein and simple sugars among children, alongside deficiencies in fats—particularly long-chain omega-3 fatty acids—as well as vitamin D and iron. Without adequate nutritional knowledge, it is difficult for parents to ensure proper dietary care for their children.

Preliminary studies have shown that even general nutrition education for parents can positively influence iron status in infants. In our study involving 200 parents, a year-long nutritional intervention beginning when the infant was six weeks old led to improvements in various iron metabolism parameters, including ferritin, transferrin, red blood cell count (RBC), hemoglobin (HGB), mean corpuscular volume (MCV), and hematocrit (HCT). Nevertheless, approximately 25% of infants in the intervention group continued to present hemoglobin levels below reference values, likely due to the general nature of the education provided. It is important to note that the intervention was not specifically designed to address iron deficiency, but rather aimed to equip parents with basic knowledge of infant nutrition [26].

In light of the widespread prevalence of iron deficiency and the well-documented efficacy of nutritional education, there is a compelling scientific rationale for the development and implementation of targeted dietary interventions aimed at enhancing iron status in infants. This constituted the principal aim of the present study.

## 2. Materials and Methods

### 2.1. Study Overview

This study was designed as a randomized controlled study [27]. The study design is presented in Figure 1. Randomization was conducted by an unblinded study assistant who was not involved in the research process. The randomization table was generated using the website randomizer.org, with an allocation ratio of 1:1. Based on the simple independent randomization scheme. The process involved the study assistant inputting the list of eligible participants into the randomizer, which then produced a sequence of assignments randomly allocating each participant to either Group A or Group B. This sequence was used to ensure an unbiased and balanced distribution of participants across groups. After randomization, participants were assigned to one of the two groups based on this pre-determined sequence. Following group assignment (Group A or B), parents received a link to download a nutrition-focused mobile application along with an activation code. Depending on which group participants were assigned to, they received the appropriate access code for the application. The app had two versions—the study group received an access code to the full version of the app, which included information on the child’s diet expansion, sample menus, and an intensive nutrition education program delivered via short text messages sent to parents approximately 4–6 times per week focused on providing practical tips and recommending age-appropriate foods. These features aimed to ensure the child receives an adequate intake of iron in their diet. In contrast, the control group only received access to a general-purpose application containing basic infant nutrition guidelines, without specific dietary advice or sample menus, they also did not receive any notifications or tailored messages. Nutritional intervention lasted until 12 months of age.

All participants provided informed consent prior to enrollment in the study. Parents were asked to read the informed consent form as well as the document concerning the collection and processing of personal data. Subsequently, they were asked to provide consent for their infant’s participation and the processing of personal data. The study received approval from the Bioethics Committee of the Poznan University of Medical Sciences (No. 394/22). The research activities were compliant with the General Data Protection Regulation (GDPR) and followed the Consolidated Standards of Reporting Trials (CONSORT).

The data obtained during the study were recorded in laboratory notebooks with specific enumeration. Missing data were double-checked during entry into the database. Sample status was blinded during the analysis. The participants’ clinical data were collected by the principal investigator and stored in password-protected.xls files. Backups were performed regularly on a monthly basis. Data security was ensured through institutional regulations at PUMS.

Parental activity was monitored using a dedicated mobile application, which tracked app usage frequency. Throughout the intervention, parents also regularly consulted with a certified dietitian to reinforce the educational message. Additionally, all participating parents completed and submitted a dietary questionnaire, confirming their commitment to the study. The readability rate of nutritional messages among parents in the intervention group was high, averaging 88% ± 25%. Notably, 70% of parents read all the nutritional messages sent during the study period, while approximately 5% read up to only 10% of the messages.

### 2.2. Participants

The study randomized 115 infants born between 36th and 42nd week of pregnancy, minimum Apgar score of eight at birth, infants less than 12 weeks of age, written consent from infant’s parent or legal guardian.

To ensure maximum comfort during participant recruitment and to include only healthy infants (without chronic conditions or seasonal infections), recruitment was primarily conducted during the first vaccination visit. As a result, 95% of infants were enrolled at 6 weeks of age, or at 8 weeks in cases of delayed appointments. The maximum age at enrollment was 12 weeks.

The exclusion criteria included: infant’s birth weight below 2500 g, history of chronic systemic disease, gastrointestinal diseases which result in digestion and absorption disorders and other severe systemic diseases (cancer, endocrinopathies, connective tissue diseases, kidney diseases, diabetes).

The patients were under the constant care of the doctor (KIB, KS, ABO) and the dietician (DW, SDC). The detailed participant flow is presented in Figure 2.

### 2.3. Intervention

The application was developed by a software engineer from the Faculty of Computing and Telecommunications at Poznan University of Technology. It was used for sending messages and included a programming interface that tracked message engagement—namely, how often messages were read and whether parents interacted with them at all. This feature enabled researchers to monitor parental engagement. The content of the messages was tailored to several factors, such as the infant’s age, developmental stage, and season. Additionally, once a month, parents were asked to indicate the infant’s feeding method (breastfed or formula-fed), and their responses further determined the type of messages they received.

All participants were recruited from family medicine clinics in the Wielkopolskie Voivodeship between 2023 and 2024.

### 2.4. Outcomes

The study’s primary endpoint was the change in hepcidin levels. Secondary endpoints comprised key biomarkers of iron metabolism: ferroportin, serum iron, transferrin, ferritin, total iron binding capacity (TIBC), unsaturated iron-binding capacity (UIBC) and high-sensitivity C-reactive protein concentration (hsCRP).

### 2.5. Blood Collection

Blood samples were collected by qualified, professional laboratory personnel. All precautionary measures were strictly followed during the collection process. A small volume of venous blood—up to 2.7 mL—was obtained from each infant at both the beginning and end of the study (S-Monovette^®^ Serum Gel CAT tubes; Sarstedt, Nümbrecht, Germany), using a clot activator for biochemical analyses. The blood samples were centrifuged at 1500× *g* for 10 min at 4 °C to separate the serum. Until the time of analysis, the samples were stored in a freezer at –80 °C. Additionally, a second blood sample was collected into an EDTA tube (S-Monovette^®^ EDTA K3E, 2.7 mL; Sarstedt, Nümbrecht, Germany).

### 2.6. Biochemical Measurements

Hemoglobin concentration was assessed using an automated hematology analyzer, Mythic^®^ 18 (Orphée, Geneva, Switzerland).

The biochemical assessment of iron status included the following parameters: serum iron concentration, ferritin, hepcidin, ferroportin, TIBC, UIBC, and hs-CRP.

Iron (ACCENT-200 FERRUM, Cormay, Cat. No. 7-258, Łomianki, Poland) and UIBC (ACCENT-200 UIBC, Cormay, Cat. No. 7-259) concentrations were determined using a colorimetric method. Transferrin (ACCENT-200 TRANSFERRIN, Cormay, Cat. No. 7-210) and ferritin (ACCENT-200 FERRITIN, Cormay, Cat. No. 7-230) concentrations were measured based on antigen–antibody reactions involving specific antibodies. All biochemical parameters were assessed using the Accent 220S automatic biochemical analyzer (Cormay, Łomianki, Poland). The sensitivity of the assays for serum iron, UIBC, transferrin, and ferritin was 4.1 µg/dL, 20 µg/dL, 0.076 g/L, and 9.1 ng/mL, respectively. The coefficients of variation (CVs) for iron measurement were 1.10% and 1.87%; 2.46% and 4.04% for UIBC; 1.12% and 4.56% for transferrin and 1.5% and 4.0% for ferritin corresponding to the repeatability and reproducibility tests. TIBC concentration was calculated by adding the measured serum iron to UIBC concentrations.

The remaining parameters were assessed using commercially available ELISA kits with reagents from Shanghai Sunred Biological Technology Co., Ltd., PRC for hepcidin (Cat. No. 201-12-1020; Sensitivity: 5.123 ng/mL; Intra-Assay CV: <10%; Inter-Assay CV: <12%) and ferroportin (Cat. No. 201-12-3769; Sensitivity: 0.175 ng/mL). High-sensitivity C-reactive protein concentration (hsCRP; Cat. No. CAN-CRP-4360; Sensitivity: 10 ng/mL; Intra-Assay CV: 15.2%; Inter-Assay CV: 9.9%) was determined using a kit from DBC Inc., London, Ontario, Canada. Spectrophotometric measurements with ELISA tests kits were made using a multi-mode microplate reader (Synergy 2 SIAFRT, BioTek, Winooski, VT, USA).

### 2.7. Body Weight Assessment

Body weight measurements were conducted by auxiliary staff (nurses). All infants were weighed using certified infant scales. Nutritional status was additionally assessed using the standardized body weight Z-score, based on cutoff points established by the World Health Organization [28]. A normal weight range was defined as a Z-score between −2 SD and +1 SD; underweight as <−2 SD to −3 SD; overweight as >+1 SD to <+2 SD; and obesity as ≥+2 SD to +3 SD.

### 2.8. Dietary Intake

At the end of the study, parents were asked to record a three-day food diary for their infant. The dietary questionnaire was based on the guidelines developed by the National Institute of Food and Nutrition [29]. Parents reported their infant’s food intake, including all meals, snacks, and fluids, along with portion sizes and times of consumption. For infants fed with formula, the amount consumed (in milliliters) was read directly from the bottle. For breastfed infants, the volume of milk intake was estimated by measuring the difference in the infant’s body weight before and after feeding. Parents were instructed to weigh their infant before and after each feeding to calculate the milk volume. Those who did not own infant scales were provided with one on loan.

They were trained and instructed by a dietitian and the attending physician on how to accurately complete the dietary records. The instructions included simple guidelines for filling out the diary correctly, as well as a link to a website that helps determine portion sizes [www.ilewazy.pl (accessed on 1 January 2024)]. Parents were able to contact the dietitian (by phone or via an app) in case of any questions.

Infant’s diets were analyzed using the Dietetyk 2015 software (Jumar Software, Poznan, Poland). The calculated average daily intake of macro- and micronutrients was compared against the recommended dietary allowances (RDA) based on Polish nutritional standards [30]. For further analysis, nutrients directly related to iron metabolism or those influencing iron absorption—such as protein, fiber, and vitamin C—were selected. When evaluating each meal, technological losses (e.g., from cooking or frying) were also taken into account.

**Figure 2 nutrients-17-03103-f002:**
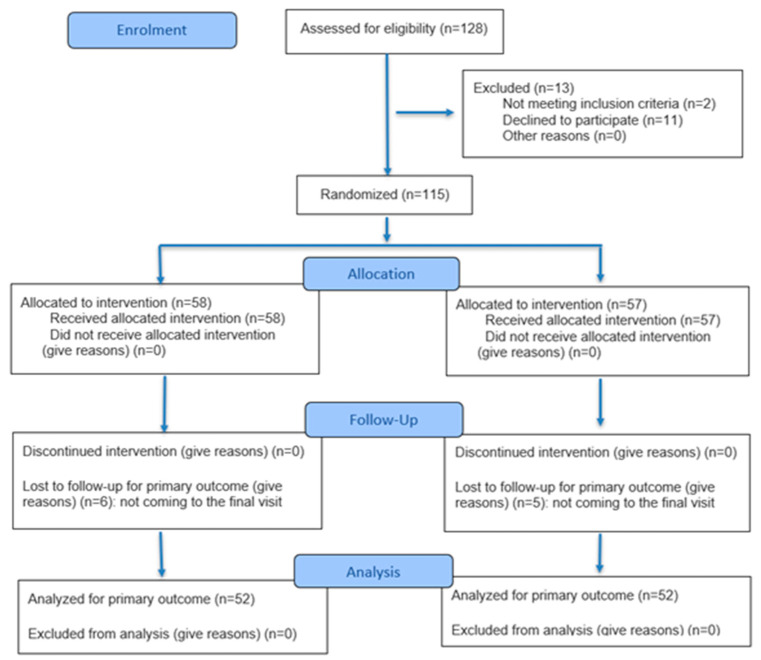
CONSORT 2025 Flow Diagram [31].

Parents were also given the opportunity to prepare a preliminary dietary record at the time of the infant’s enrollment in the study. Based on our experience, the assessment of macro- and micronutrient intake in infants during the first months of life is subject to considerable error. It is generally easier for parents to record nutritional information from jar labels and measure the quantity of formula consumed from a bottle than to accurately estimate breast milk intake using pre- and post-feeding weight measurements—particularly given that breastfeeding sessions can be lengthy during early infancy (Appendix A).

### 2.9. Minimum Sample Size

Power analyses were performed using G*Power version 3 (University of Düsseldorf, Düsseldorf, Germany). Assuming a standard deviation of 10% and a between-group difference of 7% (based on preliminary findings from a study using similar hemoglobin assessments), a sample size of 104 participants (52 per group) was calculated to achieve a statistical power of 95% (1 − β = 0.95) with a significance level of α = 0.05. To account for an anticipated 20% loss of follow-up, the final sample size was calculated to be 124 infants. Ultimately, 128 infants were assessed for eligibility, of whom 115 were successfully randomized (58 and 57 infants in each group, respectively).

### 2.10. Statistical Analysis

All results were subjected to statistical analysis. Data are presented as medians with interquartile ranges (1st–3rd quartiles), as well as means with standard deviations (SD) and 95% confidence intervals (95% CI) in accordance with the CONSORT statement. Normality of data distribution was assessed using the Shapiro–Wilk test. In cases where normality was not confirmed, the Mann–Whitney U test or χ2 test was applied to compare results between groups. To evaluate intragroup changes between pre- and post-intervention values in both study and control groups, either the paired sample *t*-test or the Wilcoxon signed-rank test was used, depending on data distribution. For variables with a normal distribution, data were analyzed using means and standard deviations. For variables with a non-normal distribution, medians and interquartile ranges were used. Additionally, to evaluate the magnitude of intra-group changes in the analyzed biochemical parameters, delta values (Δ) were calculated as the difference between post- and pre-intervention measurements. All hypotheses were tested at a significance level of α = 0.05. Statistical analyses were conducted using GraphPad Prism version 5.01 (GraphPad Software, Inc., La Jolla, CA, USA) and Statistica version 13.0 (TIBCO Software Inc., Palo Alto, CA, USA).

## 3. Results

A total of 128 individuals were assessed for eligibility, of whom 104 infants completed the study (81.2%). Eleven parents declined to enroll their infant in the study. Two infants did not meet the inclusion criteria due to a diagnosis of a chronic illness. Ultimately, 115 infants were randomized. Eleven parents failed to attend the post-study visit and did not provide a reason for withdrawal (contact with these participants was lost; a possible explanation may be that the families left the country, as all of the infants belonged to migrant families). Figure 2 presents the participant flow in accordance with CONSORT 2025 guidelines.

Anthropometric data of participants and data concerning parents’ age, education, and place of residence from both groups at the time of study enrollment are presented in Table 1. Anthropometric parameters of infants and data of parents did not differ significantly between the two groups at baseline (*p* > 0.05). The mean body weight in the study group was 5646 ± 1660 g, compared to 5841 ± 1326 g in the control group. Body length measurements were omitted during recruitment due to the high likelihood of measurement error typically associated with this parameter in infants.

Parental education levels did not differ significantly between the study groups (*p* > 0.05).

All analyzed iron metabolism parameters at the start of the study did not differ significantly between the compared groups (Table 2). Following the nutritional education intervention, the mean iron concentration was 67.7 ± 25.7 µg/dL in the study group and 58.8 ± 35.1 µg/dL in the control group. A comparison of blood test results taken one year after the educational intervention (follow up vs. follow up) showed that the study group had significantly higher levels of hemoglobin, ferritin and lower levels of TIBC and ferroportin. In both the study and control groups, ferritin levels measured at the beginning and end of the study decreased, while the levels of markers indicating iron deficiency (transferrin, UIBC, TIBC, hepcidin, and ferroportin) increased.

Additionally, intra-group changes in iron status parameters were assessed. The comparison of delta values (understood as the difference between the concentrations of individual parameters measured at the end and the beginning of the study) revealed a statistically significantly smaller increase in ferroportin concentration in the study group (Table 3).

Finally, the intake of dietary iron, as well as selected macro- and micronutrients known to influence iron absorption—such as protein, fiber, and vitamin C (Table 4)—was assessed. The results were expressed as a percentage of RDA fulfilment. Following one year of nutritional education, infants in the study group demonstrated significantly higher consumption of both iron and vitamin C. The median iron intake in the study group was 96.1% of the RDA, compared to 58.6% in the control group. In contrast, protein and fiber intake remained comparable between the study and control groups. Of particular concern was the excessively high protein intake observed in both groups, exceeding 250% of the RDA.

## 4. Discussion

Our study demonstrated that parental nutritional education positively influences iron metabolism in infants. Infants in the study group not only consumed higher amounts of iron and vitamin C, but also exhibited higher hemoglobin and serum ferritin concentrations and lower levels of iron deficiency markers (TIBC and ferroportin). Additionally, the mean increase in ferroportin concentration (expressed as delta value) was lower in the study group.

According to current knowledge, anemia in children remains one of the most serious health problems worldwide. The greatest burden occurs in regions with low socio-demographic indicators, although it is also a significant clinical problem in developed countries [11]. Nutritional deficiencies, particularly iron deficiency, are among the main causes of anemia. Iron deficiency leads to cognitive impairment, weakened immunity, and developmental delays in children. Therefore, early detection and implementation of effective preventive strategies are crucial to reducing the incidence of anemia [32,33].

Recent studies have demonstrated that intensive nutritional education can significantly improve health outcomes in infants [34,35,36,37]. However, studies examining the impact of nutritional education on iron status in infants are limited. Most of these studies are based on patients with anaemia or partially include iron and/or other nutrient supplementation [36,38,39].

Improving iron status in infants through nutritional education was primarily intended to increase dietary intake of iron and components that contribute to its absorption and assimilation, i.e., vitamin C, dietary fiber, and proteins [40,41,42]. Vitamin C plays an important role, increasing the absorption of non-heme iron in the small intestine and improving the iron status in the body [40].

Statistically significant differences in iron and vitamin C intake were observed among infants whose parents received intensive nutritional education, as compared to those in the control group (*p* < 0.05). Furthermore, iron intake in the study group was generally adequate following nutritional education. The median percentage of the recommended daily iron intake met in the study group was 96.1%, compared to only 58.6% in the control group. This underscores the substantial impact of the nutritional program.

Vitamin C intake, statistically significant differences were also noted between the control and study groups, although the intake levels in both groups fell within the recommended daily range.

Protein and dietary fiber intake did not differ significantly between the study groups. The average fiber intake in both groups was within the normal range. Adequate fiber consumption is extremely important. While normal fiber intake does not interfere with iron absorption [43], excessive fiber consumption—often associated with high dietary levels of phytic acid—can significantly reduce iron bioavailability. Phytates may decrease iron absorption by up to 50% [44]. On the other hand, the benefits of dietary fiber are increasingly emphasized. Early exposure to fiber-rich foods may support long-term gastrointestinal health and the development of healthy eating habits. Dietary fiber contributes to digestive health by promoting the growth of beneficial gut microbiota, which colonize the intestines most intensively during infancy. Acting as a prebiotic, fiber nourishes bacteria of the genus Lactobacillus (e.g., *Lactobacillus plantarum* 299v), which enhance nutrient availability for intestinal epithelial cells and other commensal bacteria [45].

Our study also assessed protein intake in the infants’ diet. The total protein intake was excessive in both groups—nearly three times higher than the recommended daily allowance. Although excessive protein intake is concerning, our findings are consistent with European studies showing that protein consumption among infants often exceeds recommended levels [15,26,46,47,48,49]. In our study, protein intake was primarily evaluated in the context of its role as a source of heme iron. While promoting meat consumption in infancy may contribute to protein overconsumption, studies assessing the impact of meat intake on infant growth have yielded mixed results [50].

Despite the increased iron and vitamin C intake in the study group, no differences in serum iron concentration were observed among the infants. There was no statistically significant difference in serum iron levels between the study and control groups, nor within the same group when comparing concentrations before and after intensive nutritional education (*p* > 0.05). This may be due to several factors. Firstly, iron concentrations in most infants were within the normal range [51]. Secondly, the educational program was directed at infants without consideration of the occurrence of anaemia. Additionally, this may be related to sufficient iron concentrations present at birth. It is also worth noting that serum iron concentration is not considered a key biochemical marker of iron status in the body. Serum ferritin concentration is currently regarded as the most reliable and specific indicator of iron availability [52]. The World Health Organization recommends using ferritin concentrations to monitor and evaluate iron-related interventions, emphasizing ferritin over serum iron because it reflects total body iron stores [53].

Ferritin plays an essential role in regulating iron metabolism and maintaining the body’s metabolic balance. After birth, infants exhibit high physiological concentrations of ferritin, as confirmed in our study. These levels gradually decreased, a trend that was also observed (*p* < 0.0001). Furthermore, a statistically significant difference in final ferritin concentrations between the control and study groups (*p* < 0.05) indicated improved iron status and validated the effectiveness of the educational program.

Inflammatory parameters should be considered when interpreting ferritin levels, as ferritin is an acute-phase reactant. Ferritin levels may remain elevated for weeks during and after episodes of infection or inflammation [54]. Therefore, hsCRP was included in our study, and no statistically significant differences were observed between the study groups (*p* > 0.05).

No statistically significant differences in transferrin and hepcidin concentrations or UIBC levels were observed between the study groups; however, both groups exhibited significant intragroup changes between baseline and follow-up measurements.

TIBC reflects the maximum amount of iron that can be bound by transferrin [55], with elevated values typically observed under iron-deficient conditions [56]. The study demonstrated statistically significant higher TIBC values in the control group compared to the intensive nutrition education group.

Ferroportin is the only known iron exporter and plays a crucial role in transporting iron from cells into the bloodstream [57]. Significant differences in ferroportin concentrations were observed between the study groups (*p* < 0.05). Moreover, statistically significant differences in the changes (delta values) of ferroportin levels were also noted between the study and control groups. It is important to emphasize that ferroportin concentration is not routinely used as a biomarker of iron metabolism. This is primarily because its functional activity is tightly regulated by hepcidin levels, which modulate ferroportin activity [58].

Ferritin and TIBC are sensitive indicators of iron status. Moreover, our study demonstrated significant differences in ferroportin concentrations between the study groups. Consequently, ferritin, TIBC, ferroportin, and hepcidin should be considered reliable biomarkers for assessing iron metabolism in infants. Additionally, it has been suggested that these parameters may provide a more sensitive measure of the effectiveness of nutritional interventions targeting iron status than serum iron concentration alone.

The study’s limitations include a relatively small sample size, which restricts the generalizability of findings and limits the potential for population-level inference regarding the impact of nutritional education. Additional methodological constraints may involve reliance on parent-reported data, which introduces potential reporting bias, as well as the absence of blinding, which could affect the objectivity of outcome assessment. Additionally, in future studies with larger cohorts and more frequent measurements, it is worth using more advanced statistical models (e.g., repeated measures ANOVA), which may provide additional information, especially in terms of baseline variability and time-dependent effects.

## 5. Conclusions

Targeted parental nutrition education can improve iron metabolism in infants. After a year of intervention, the study group demonstrated not only higher iron intake but also increased ferritin levels and decreased concentrations of parameters typically associated with increased iron absorption—namely, total iron binding capacity (TIBC) and ferroportin. These findings warrant future studies involving larger cohorts, extended follow-up periods, and comprehensive assessments of infants development.

## Figures and Tables

**Figure 1 nutrients-17-03103-f001:**
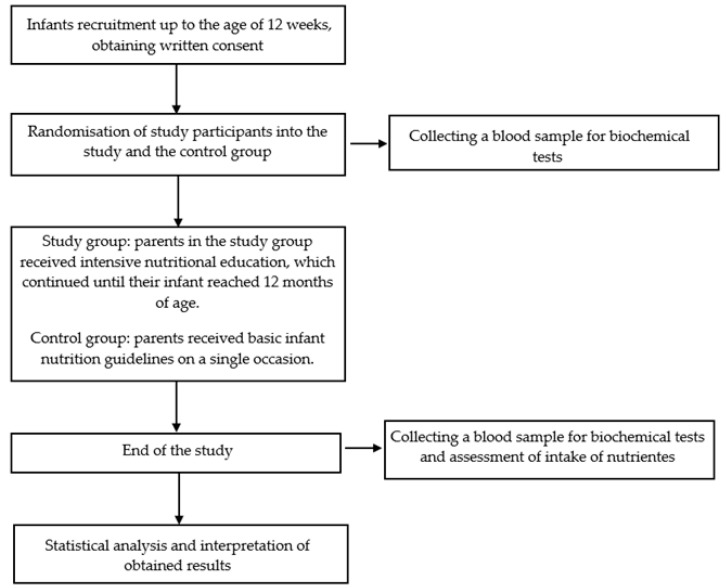
Scheme of the study.

**Table 1 nutrients-17-03103-t001:** Anthropometric data of infants and data concerning parents’ age, education, and place of residence at baseline.

Parameter	Study Group		Control Group		*p*
	(n = 58)		(n = 57)		
	Median	Mean ± SD	Median	Mean ± SD	
	(Q1–Q3)	(95% CI)	(Q1–Q3)	(95% CI)	
Infants					
Body weight [g]	5280	5646 ± 1660	6175	5841 ± 1326	0.3105
	(4305–7230)	(4992–6300)	(4810–6795)	(5471–6428)	
*Z*-score for body weight at baseline	−0.12	0.59 ± 2.53	0.26	0.23 ± 1.39	0.6851
	(−1.19–2.88)	(−0.45–1.47)	(−0.43–1.08)	(−0.27–0.78)	
Sex: female	50.00%		45.60%		0.6415
Parents					
Place of residence					
Village (from the city agglomeration)	61%		57%		
a city with fewer than 500,000 residents	24%		28%		0.7248
a city with more than 500,000 residents	15%		15%		
Education					
Primary	2%		2%		
Secondary	25%		22%		0.6983
Higher	73%		76%		

Q—quartile, SD—standard deviation, CI—confidence interval. *p*-value for the U Mann-Whitney test or the χ2 test.

**Table 2 nutrients-17-03103-t002:** Comparison of measured parameters between the study group and the control group.

Parameters	Baseline					Follow-Up					*p* ^I^	*p* ^II^
	Study Group (n = 52)		Control Group (n = 52)		*p*	Study Group (n = 52)		Control Group (n = 52)		*p*		
	Median (Q1–Q3)	Mean ± SD (95% CI)	Median (Q1–Q3)	Mean ± SD (95% CI)		Median (Q1–Q3)	Mean ± SD (95% CI)	Median (Q1–Q3)	Mean ± SD (95% CI)			
Hemoglobin (g/dL)	12.0 (11.2–12.7)	12.0 ± 1.6 (11.4–12.6)	11.9 (11.1–12.8)	11.9 ± 1.3 (11.4–12.4)	0.7834 ^1^	12.0 (11.0–12.8)	11.9 ± 0.9 (11.6–12.8)	11.2 (10.6–12.1)	11.5 ± 1.2 (11.0–11.9)	**0.0499 ^2^**	0.6811 ^3^	**0.0165 ^3^**
Iron (µg/dL)	76.3 (51.9–108.4)	77.9 ± 40.0 (63.5–92.3)	58.5 (44.5–75.5)	65.8 ± 31.4 (54.5–77.2)	0.1951 ^1^	65.0 (52.6–87.0)	67.7 ± 25.7 (58.1–77.3)	55.6 (37.1–70.9)	58.8 ± 35.1 (46.1–71.5)	0.1038 ^1^	0.0840 ^3^	0.0788 ^3^
Ferritin (ng/mL)	175.5 (71.9–274.8)	207.0 ± 169.0 (146.1–267.9)	151.3 (77.5–219.1)	170.3 ± 99.1 (134.6–206.1)	0.7119 ^1^	35.1 (27.9–42.4)	46.3 ± 53.3 (26.4–66.3)	18.9 (14.9–37.6)	29.1 ± 21.1 (21.5–36.7)	**0.0067 ^1^**	**<0.0001** ^3^	**<0.0001** ^3^
Transferrin (g/L)	2.07 (1.77–2.51)	2.18 ± 0.59 (1.97–2.39)	2.33 (2.09–2.65)	2.38 ± 0.48 (2.20–2.55)	0.1131 ^2^	3.09 (2.82–3.31)	3.06 ± 0.51 (2.87–3.25)	3.02 (2.69–3.25)	3.01 ± 0.46 (2.84–3.17)	0.4681 ^2^	**<0.0001** ^2^	**<0.0001** ^2^
UIBC (µg/dL)	169.0 (136.5–265.7)	196.3± 80.1 (167.5–225.2)	222.0 (176.5–261.0)	225.4 ± 70.1 (199.9–250.9)	0.1470 ^2^	286.0 (259.7–309.7)	280.4± 37.9 (266.2–294.5)	300.0 (264.5–331.0)	287.6 ± 55.4 (267.6–307.6)	0.2480 ^1^	**<0.0001** ^3^	**0.0003** ^3^
TIBC (µg/dL)	259.0 (226.0–324.2)	272.4 ± 61.9 (250.1–294.8)	285.0 (257.5–304.5)	285.8 ± 55.8 (265.7–305.9)	0.2797 ^2^	356.5 (330.5–379.5)	347.1 ± 45.4 (330.8–364.7)	382.0 (293.5–406.5)	361.5 ± 68.7 (336.7–386.2)	**0.0478** ^1^	**<0.0001** ^3^	**<0.0001** ^3^
Hepcidin (ng/mL)	244.0 (132.6–444.2)	395.9 ± 386.1 (256.7–535.1)	339.3 (239.9–521.0)	416.8 ± 259.9 (323.1–510.5)	0.1086 ^1^	729.7 (379.4–902.8)	649.3 ± 338.5 (520.6–778.1)	699.5 (422.7–1076.1)	727.7 ± 431.5 (572.1–883.2)	0.6388 ^2^	**0.0031** ^3^	**0.0004** ^3^
Ferroportin (ng/mL)	9.2 (6.4–15.6)	15.6 ± 15.4 (10.0–21.1)	12.7 (9.7–220.9)	16.0 ± 9.5 (12.6–19.4)	0.0820 ^1^	22.8 (15.3–28.5)	22.9 ± 11.8 (18.4–27.4)	32.8 (17.5–50.3)	34.4 ± 20.9 (26.9–41.9)	**0.0410** ^1^	**0.0107** ^3^	**0.0002** ^3^
hsCRP (ng/mL)	9.7 (2.8–30.1)	32.0 ± 54.9 (12.3–51.9)	15.6 (3.6–35.0)	26.3 ± 28.9 (15.9–36.7)	0.5101 ^1^	7.2 (2.2–16.1)	35.8 ± 76.4 (6.8–64.9)	17.5 (7.4–40.3)	24.0 ± 24.6 (15.1–32.9)	0.2599 ^1^	0.2893 ^3^	0.6469 ^3^

Q—quartile, SD—standard deviation, CI—confidence interval; UIBC—unsaturated iron-binding capacity, TIBC—total iron binding capacity, hsCRP—high-sensitivity C-reactive protein; ^I^ Study group: baseline vs. follow-up; ^II^ Control group: baseline vs. follow-up; ^1^ Mann–Whitney U test; ^2^ Student’s *t*-test; ^3^ Wilcoxon signed-rank test. Statistically significant differences have been highlighted in bold.

**Table 3 nutrients-17-03103-t003:** Comparison of measured parameters between the study group and the control group at the end of the trial—delta versus delta.

Parameters	Study Group (n = 52)		Control Group (n = 52)		*p*
	Δ Follow-Up—Baseline		Δ Follow-Up—Baseline		
	Median (Q1–Q3)	Mean ± SD (95% CI)	Median (Q1–Q3)	Mean ± SD (95% CI)	
Δ Hemoglobin (g/dL)	0.0 (−0.7–0.9)	−0.1 ± 1.4 (−0.6–0.4)	−0.4 (−1.2–0.1)	−0.4 ± 1.1 (−0.8–−0.1)	0.1000 ^1^
Δ Iron (µg/dL)	−13.5 (−34.4–0.0)	−14.5 ± 48.2 (−31.8–2.9)	−1.4 (−18.1–1.5)	−7.0 ± 23.9 (−15.6–1.6)	0.1397 ^1^
Δ Ferritin (ng/mL)	−138.0 (−223.7–−30.3)	−163.5 ± 174.6 (−226.5–−100.5)	−137.1 (−201.3–−52.8)	−141.3 ± 89.8 (−173.7–−108.9)	0.8351 ^1^
Δ Transferrin (g/L)	0.75 (0.27–1.33)	0.69 ± 0.82 (0.39–0.99)	0.44 (0.34–1.14)	0.63 ± 0.61 (0.41–0.85)	0.4766 ^2^
Δ UIBC (µg/dL)	62.0 (3.0–144.5)	66.5 ± 86.9 (35.2–97.8)	54.0 (7.5–108.0)	62.2 ± 80.1 (33.1–91.2)	0.8932 ^1^
Δ TIBC (µg/dL)	67.0 (0.25–118.0)	53.5 ± 91.0 (21.7–87.5)	96.0 (15.5–128.0)	75.7 ± 76.8 (48.0–103.4)	0.4089 ^1^
Δ Hepcidin (ng/mL)	218.2 (−124.7–505.9)	192.6 ± 410.1 (44.7–340.4)	210.8 (61.7–626.9)	310.8 ± 464.1 (143.6–478.2)	0.3507 ^2^
Δ Ferroportin (ng/mL)	6.7 (−2.5–17.0)	5.2 ± 16.1 (−0.6–11.0)	17.4 (0.8–1.1)	18.4 ± 22.2 (10.4–26.4)	**0.0237** ^2^
Δ hsCRP (ng/mL)	−1.3 (−23.9–1.9)	0.4 ± 95.7 (−34.1–34.9)	4.5 (−7.6–12.9)	−2.3 ± 36.7 (−15.4–10.8)	0.1527 ^1^

Q—quartile, SD—standard deviation, CI—confidence interval; UIBC –unsaturated iron-binding capacity, TIBC—total iron binding capacity, hsCRP—high-sensitivity C-reactive protein; ^1^ Mann–Whitney U test; ^2^ Student’s *t*-test. Statistically significant differences have been highlighted in bold.

**Table 4 nutrients-17-03103-t004:** Comparison of dietary intake between the study group and the control group at the end of the trial.

Dietary Intake **(% RDA)**	Study Group (n = 52)		Control Group (n = 52)		*p*
	Follow-Up		Follow-Up		
	Median (Q1–Q3)	Mean ± SD (95% CI)	Median (Q1–Q3)	Mean ± SD (95% CI)	
Iron	96.1 (78.3–124.3)	102.6 ± 48.2 (77.7–127.4)	58.6 (44.3–88.6)	77.6 ± 65.0 (44.2–111.1)	**0.0252** ^1^
Proteins	305.6 (271.1–343.4)	314.6 ± 73.3 (276.9–352.3)	271.9 (233.8–291.5)	279.6 ± 58.0 (249.8–309.5)	0.1386 ^1^
Fiber	106.3 (87.1–145.0)	122.6 ± 62.5 (90.4–154.7)	102.0 (29.5–170.0)	118.5 ± 86.7 (73.9–163.1)	0.5816 ^2^
Vitamin C	195.4 (149.9–312.4)	260.0 ± 171.8 (171.7–348.4)	144.8 (110.0–232.5)	161.8 ± 77.0 (122.3–201.4)	**0.0458** ^1^

RDA—Recommended Dietary Allowance according to Polish nutritional standard; ^1^ Mann–Whitney U test; ^2^ Student’s *t*-test. Statistically significant differences have been highlighted in bold.

## Data Availability

The datasets generated and analyzed during the current study are available from the corresponding author upon reasonable request.

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
