# Peer review of "The Influence of Intensive Nutritional Education on the Iron Status in Infants—Randomised Controlled Study"

_nutrients, 2025, doi:10.3390/nu17193103_

Round 1
Reviewer 1 Report
Comments and Suggestions for Authors
I had the great privilege to review the manuscript entitled “The influence of intensive nutritional education on the iron sta- 2 tus in infants - randomised controlled study”. However, the manuscript can be improved by addressing the following points that the authors should consider.
1. The introduction provides a basic background but lacks a broader global perspective. The authors mainly cite data from Europe and developed countries while neglecting the prevalence and burden of infant iron deficiency or anemia in Asia, Africa, and the Americas. Including such data would better highlight the global significance of the issue and strengthen the study rationale.
2. Details of randomization and blinding procedures are insufficient, and the intervention content and monitoring are inadequately described. The reliability of dietary assessment (e.g., food diary validation, estimation of breast milk intake) is not well addressed. The justification of sample size calculation and statistical analysis also requires clearer explanation.
3. Although the main findings are presented, the participant flow is described too briefly, with inadequate explanation of attrition. Anthropometric data are unnecessarily repeated, and the biological significance of observed changes is underexplained. The tables are overly dense and lack sufficient narrative interpretation. Clearer linkage of statistical results to clinical relevance would enhance this section.
4. The discussion highlights key findings but remains limited in scope. Results are not sufficiently placed in a global context, nor compared with similar interventions in other regions. Mechanisms by which parental education improves iron metabolism are only briefly mentioned, without deeper integration of existing evidence. Study limitations (e.g., attrition, reliance on parental reports, lack of blinding) are underemphasized.
5. While concise, the conclusions are somewhat overstated. They emphasize the positive impact of parental nutritional education without adequately acknowledging limitations such as small sample size, regional specificity, and lack of long-term follow-up. The authors should temper their claims, clarify the need for confirmation in larger and more diverse populations, and highlight the practical implications for policy, clinical practice, and public health nutrition.
6. Ethics and transparency: Although ethical approval and informed consent are mentioned, the manuscript does not provide details on data management, handling of missing data, or registration in a clinical trial database. Adding these details would strengthen research transparency.
Author Response
Response to the comments made by the reviewers
Manuscript ID: nutrients-3836723
Title: The influence of intensive nutritional education on the iron status in infants - randomised controlled study
We would like to thank the Reviewers for their careful review of our manuscript and for providing suggestions to improve its quality. We have carried out a significant revision of the manuscript and we believe the paper has been significantly improved.
According to the Reviewers’ suggestion, the manuscript has been carefully checked and corrected. The changes in the manuscript have been highlighted in red.
Below we sequentially address all of the points raised by the Reviewers.
Reviewer 1:
Comments and Suggestions for Authors
- The introduction provides a basic background but lacks a broader global perspective. The authors mainly cite data from Europe and developed countries while neglecting the prevalence and burden of infant iron deficiency or anemia in Asia, Africa, and the Americas. Including such data would better highlight the global significance of the issue and strengthen the study rationale.
Response: This section has been expanded with the following:
According to the Global Burden of Disease Study (2019), anemia imposed a substantial global burden on children under five, with a reported years lived with disability of 1253 (95% UI: 831-1831) per 100000 population [11].
In Europe, the prevalence of iron deficiency among infants aged 6 to 12 months is estimated to range from 4% to 18% [12]. Among toddlers over 12 months of age living in developed countries (such as the United States, New Zealand, Finland, and Greece), the prevalence of iron deficiency without anemia may reach up to 30%) [13–16].Globally, the issue of iron deficiency and the resulting anemia is even more widespread. According to WHO data, the global prevalence of IDA among preschool-aged children (0–4.99 years) is 42.6% (95% CI: 37.7–47.4). The highest prevalence was recorded in Africa, at 62.3% (95% CI: 59.6–64.8). Among all African countries, anemia was most prevalent in Burkina Faso, where the rate reached 86% (95% CI: 82–89). High rates of anemia were also observed in South-East Asia, where the prevalence of IDA was 53.8% (95% CI: 39.9–63.9), including as much as 61% (95% CI: 51–69) in Pakistan [17,18].
- Details of randomization and blinding procedures are insufficient, and the intervention content and monitoring are inadequately described. The reliability of dietary assessment (e.g., food diary validation, estimation of breast milk intake) is not well addressed. The justification of sample size calculation and statistical analysis also requires clearer explanation.
Response: We edited the manuscript, and the following explanation has been added.
- Randomization
This study was designed as a randomized controlled study [27]. The study design is presented in Figure 1. Randomization was conducted by an unblinded study assistant who was not involved in the research process. The randomization table was generated using the website randomizer.org, with an allocation ratio of 1:1. Based on the simple independent randomization scheme. The process involved the study assistant inputting the list of eligible participants into the randomizer, which then produced a sequence of assignments randomly allocating each participant to either Group A or Group B. This sequence was used to ensure an unbiased and balanced distribution of participants across groups. After randomization, participants were assigned to one of the two groups based on this pre-determined sequence. Following group assignment (Group A or B), parents received a link to download a nutrition-focused mobile application along with an activation code. Depending on which group participants were assigned to, they received the appropriate access code for the application. The app had two versions – the study group received an access code to the full version of the app, which included information on the child's diet expansion, sample menus, and an intensive nutrition education program delivered via short text messages sent to parents approximately 4–6 times per week focused on providing practical tips and recommending age-appropriate foods. These features aimed to ensure the child receives an adequate intake of iron in their diet. In contrast, the control group only received access to a general-purpose application containing basic infant nutrition guidelines, without specific dietary advice or sample menus, they also did not receive any notifications or tailored messages. The nutritional intervention lasted until the infant reached 12 months of age.
- Participants
The study randomized 115 infants born between 36th and 42nd week of pregnancy, minimum Apgar score of eight at birth, infants less than 12 weeks of age, written consent from infant’s parent or legal guardian.
The exclusion criteria included: infants’ birth weight below 2500g, history of chronic systemic disease, gastrointestinal diseases which result in digestion and absorption disorders and other severe systemic diseases (cancer, endocrinopathies, connective tissue diseases, kidney diseases, diabetes).
The patients were under the constant care of the doctor (KIB, KS, ABO) and the dietician (DW, SDC). The detailed participant flow is presented in Figure 2.
- Food diary validation
At the end of the study, parents were asked to record a three-day food diary for their infant. The dietary questionnaire was based on the guidelines developed by the National Institute of Food and Nutrition [29]. Parents reported their infant’s food intake, including all meals, snacks, and fluids, along with portion sizes and times of consumption. For infants fed with formula, the amount consumed (in milliliters) was read directly from the bottle. For breastfed infants, the volume of milk intake was estimated by measuring the difference in the infant’s body weight before and after feeding. Parents were instructed to weigh their infant before and after each feeding to calculate the milk volume. Those who did not own infant scales were provided with one on loan.
- Minimum sample size - The minimum sample size was maintained for hemoglobin in accordance with the design assumptions.
Power analyses were performed using G*Power version 3 (University of Düsseldorf, Düsseldorf, Germany). Assuming a standard deviation of 10% and a between-group difference of 7% (based on preliminary findings from a study using similar hemoglobin assessments), a sample size of 104 participants (52 per group) was calculated to achieve a statistical power of 95% (1–β = 0.95) with a significance level of α = 0.05. To account for an anticipated 20% loss of follow-up, the final sample size was calculated to be 124 infants. Ultimately, 128 infants were assessed for eligibility, of whom 115 were successfully randomized (58 and 57 infants in each group, respectively).
- Statistical analysis
All results were subjected to statistical analysis. Data are presented as medians with interquartile ranges (1st–3rd quartiles), as well as means with standard deviations (SD) and 95% confidence intervals (95% CI) in accordance with the CONSORT statement. Normality of data distribution was assessed using the Shapiro–Wilk test. In cases where normality was not confirmed, the Mann–Whitney U test or χ2 test was applied to compare results between groups. To evaluate intragroup changes between pre- and post-intervention values in both study and control groups, either the paired sample t-test or the Wilcoxon signed-rank test was used, depending on data distribution. For variables with a normal distribution, data were analyzed using means and standard deviations. For variables with a non-normal distribution, medians and interquartile ranges were used. Additionally, to evaluate the magnitude of intra-group changes in the analyzed biochemical parameters, delta values (Δ) were calculated as the difference between post- and pre-intervention measurements. All hypotheses were tested at a significance level of α = 0.05. Statistical analyses were conducted using GraphPad Prism version 5.01 (GraphPad Software, Inc., La Jolla, CA, USA) and Statistica version 13.0 (TIBCO Software Inc., Palo Alto, CA, USA).
- 3. Although the main findings are presented, the participant flow is described too briefly, with inadequate explanation of attrition. Anthropometric data are unnecessarily repeated, and the biological significance of observed changes is underexplained. The tables are overly dense and lack sufficient narrative interpretation. Clearer linkage of statistical results to clinical relevance would enhance this section.
Response: Missing information regarding participant flow has been added, and duplicate content has been removed. The narrative interpretation has been expanded. The tables present data on the median, mean, standard deviation, quartiles, and 95% confidence intervals, in accordance with CONSORT guidelines. Splitting the tables into smaller units would have led to data redundancy; therefore, all relevant information has been consolidated into a single table.
The study randomized 115 infants born between 36th and 42nd week of pregnancy, minimum Apgar score of eight at birth, infants less than 12 weeks of age, written consent from infant’s parent or legal guardian.
The exclusion criteria included: infants’s birth weight below 2500g, history of chronic systemic disease, gastrointestinal diseases which result in digestion and absorption disorders and other severe systemic diseases (cancer, endocrinopathies, connective tissue diseases, kidney diseases, diabetes).
The patients were under the constant care of the doctor (KIB, KS, ABO) and the dietician (DW, SDC). The detailed participant flow is presented in Figure 2.
- The discussion highlights key findings but remains limited in scope. Results are not sufficiently placed in a global context, nor compared with similar interventions in other regions. Mechanisms by which parental education improves iron metabolism are only briefly mentioned, without deeper integration of existing evidence. Study limitations (e.g., attrition, reliance on parental reports, lack of blinding) are underemphasized.
Response: This section has been expanded with the following:
Our study demonstrated that parental nutritional education positively influences iron metabolism in infants. Infants in the study group not only consumed higher amounts of iron and vitamin C, but also exhibited higher serum ferritin concentrations and lower levels of iron deficiency markers (TIBC and ferroportin). Additionally, the mean increase in ferroportin concentration (expressed as delta value) was lower in the study group.
According to current knowledge, anemia in children remains one of the most serious health problems worldwide. The greatest burden occurs in regions with low socio-demographic indicators, although it is also a significant clinical problem in developed countries [11]. Nutritional deficiencies, particularly iron deficiency, are among the main causes of anemia. Iron deficiency leads to cognitive impairment, weakened immunity, and developmental delays in children. Therefore, early detection and implementation of effective preventive strategies are crucial to reducing the incidence of anemia [32,33].
Protein and dietary fiber intake did not differ significantly between the study groups. The average fiber intake in both groups was within the normal range. Adequate fiber consumption is extremely important. While normal fiber intake does not interfere with iron absorption [43], excessive fiber consumption—often associated with high dietary levels of phytic acid—can significantly reduce iron bioavailability. Phytates may decrease iron absorption by up to 50% [44]. On the other hand, the benefits of dietary fiber are increasingly emphasized. Early exposure to fiber-rich foods may support long-term gastrointestinal health and the development of healthy eating habits. Dietary fiber contributes to digestive health by promoting the growth of beneficial gut microbiota, which colonize the intestines most intensively during infancy. Acting as a prebiotic, fiber nourishes bacteria of the genus Lactobacillus (e.g., Lactobacillus plantarum 299v), which enhance nutrient availability for intestinal epithelial cells and other commensal bacteria [45].
Our study also assessed protein intake in the infants’ diet. The total protein intake was excessive in both groups—nearly three times higher than the recommended daily allowance. Although excessive protein intake is concerning, our findings are consistent with European studies showing that protein consumption among infants often exceeds recommended levels [15,26,46–49]. In our study, protein intake was primarily evaluated in the context of its role as a source of heme iron. While promoting meat consumption in infancy may contribute to protein overconsumption, studies assessing the impact of meat intake on infant growth have yielded mixed results [50].
The study's limitations include a relatively small sample size, which restricts the generalizability of findings and limits the potential for population-level inference regarding the impact of nutritional education. Additional methodological constraints may involve reliance on parent-reported data, which introduces potential reporting bias, as well as the absence of blinding, which could affect the objectivity of outcome assessment.
- While concise, the conclusions are somewhat overstated. They emphasize the positive impact of parental nutritional education without adequately acknowledging limitations such as small sample size, regional specificity, and lack of long-term follow-up. The authors should temper their claims, clarify the need for confirmation in larger and more diverse populations, and highlight the practical implications for policy, clinical practice, and public health nutrition.
Response: The conclusions were corrected:
Targeted parental nutrition education can improve iron metabolism in infants. After a year of intervention, the study group demonstrated not only higher iron intake but also increased ferritin levels and decreased concentrations of parameters typically associated with increased iron absorption—namely, total iron binding capacity (TIBC) and ferroportin. These findings warrant future studies involving larger cohorts, extended follow-up periods, and comprehensive assessments of infants development.
- Ethics and transparency: Although ethical approval and informed consent are mentioned, the manuscript does not provide details on data management, handling of missing data, or registration in a clinical trial database. Adding these details would strengthen research transparency.
Response: Thank you for highlighting this important aspect. We have now added details regarding data management and the handling of missing data in the section - 2.1. Study overview.
The data obtained during the study were recorded in laboratory notebooks with specific enumeration. Missing data were double-checked during entry into the database. Sample status was blinded during the analysis. The participants’ clinical data were collected by the principal investigator and stored in password-protected.xls files. Backups were performed regularly on a monthly basis. Data security was ensured through institutional regulations at PUMS.
Unfortunately, the study was not registered in a clinical trial database. However, we acknowledge the importance of such registration and will ensure compliance in future studies.
Reviewer 2 Report
Comments and Suggestions for Authors
Dear Authors:
Regarding the manuscript with title “The influence of intensive nutritional education on the iron status in infants - randomised controlled study”, I have major concerns. Also several minor comments were addressed.
Major Concerns:
Comment 1: Regarding the number of participants, the data on Abstract (200 participants), Methods (62 children in each subgroup) and Figure 2 (128 participants) is contradictory.
Comment 2: Why authors only presente anthropometric data for the characterization of the sample? Information regarding age, sex and place of residence will be crucial for readers.
Comment 3: Lines 215-217: “A comparison of blood test results taken one year after the educational intervention showed that the experimental group had significantly higher levels of ferritin and lower levels of TIBC and ferroportin”. This information is not in accordance with results presented on Table 2.
Comment 4: A single repeated measures analysis (RM-ANOVA) or a Mixed Models approach would be more appropriate, powerful, and informative than calculating multiple point-wise differences and conducting separate comparisons (Table 2 and 3). Also, since the study includes both pre- and post-test measurements, comparing dietary intake only at the end of the trial is misleading. Differences at baseline are not accounted for, and the analysis does not assess changes over time (Table 4).
Comment 5: It is crucial that authors presente data regarding the Parental engagement, as it influences the results of the outcomes.
Comment 6: In the tables, both mean and median values are presented together with a single p-value. This may be confusing, since the statistical test applied (and therefore the p-value) should correspond to the main measure of central tendency chosen, depending on the distribution of the data. It would be clearer to either: a) select one primary measure (mean or median, depending on data distribution) and present the p-value from the appropriate test or (b) present both descriptive measures but specify explicitly which one guided the statistical comparison."
Comment 7: What are the main differences between this study and the already published study (reference 19)?
Minor comments:
Comment 1: Line 21: Authors must change “and the control group without any intervention.” by “and the control group, which received basic infant nutrition guidelines
Comment 2: Lines 19-23: Authors must clarify the methods used to analyse the intake of nutrients and biochemical parameters.
Comment 3: Line 26: “TIBC value”. Authors must add the meaning of this abbreviation.
Commen 4: On Abstract, authors must refer the direction of the results presented. The way i tis written do not allow readers to understand which group obtained the better results.
Comment 5: Lines 40-41: It will be important that authors presente results beyond the ages of 6 to 12 months.
Comment 6: Line 81 and Figure 1: “infants less than 8 weeks of age,” and “up to age of 12 weeks”. There is a discrepancy in this inclusion criteria. I kindly ask authors to correct it.
Comment 7: Lines 96-97: Authors must insert the exact duration of the intervention. (in the case “12 months”)
Comment 8: On Figure 1: authors must refer the exact duration of the intervention and not present na interval between 11 and 13 months.
Comment 9: On Figure 1, authors must also presente information regarding the control group (intervention and number of participants)
Comment 10: On outcomes, authors must insert the “intake of nutrientes”.
Comment 11: There is a discrepancy in the outcomes presented and the outcomes presented on biochemical measurements.
Comment 12: Line 156. Not only at the end of the study but also in the beginning, correct?
Comment 13: Lines 177-178: Accordingly, 62 children will be recruited and randomly assigned to each subgroup.” This is not a protocol, this is not a study that will be performed. In the presente study 200 participantes were included in the presente study. I kindly ask authord to check this sentence.
Comment 14: On Figure 2, authors must change “Excluded (N= 128)” by “Excluded (N=13)”.
Comment 15: Lines 192-193. Authors refer to 8 parents that failed to attend the post-study. However on Figure 2, information stated 11 parents (6 from intervention and 5 from control group)
Comment 16: Authors must uniformize the terminology (experimental group OR study group) and (infants VS. young children). In the first case, authors can choose between the two valid options. In the second case, regarding the scope of the study, infants will be more correct.
Comment 17: On the second paragraph of Discussion, authors must also refer the importance of dietary fibers and proteins.
Comment 18: In the beginning of Discussion, authors must presente the main results.
Author Response
Response to the comments made by the reviewers
Manuscript ID: nutrients-3836723
Title: The influence of intensive nutritional education on the iron status in infants - randomised controlled study
We would like to thank the Reviewers for their careful review of our manuscript and for providing suggestions to improve its quality. We have carried out a significant revision of the manuscript and we believe the paper has been significantly improved.
According to the Reviewers’ suggestion, the manuscript has been carefully checked and corrected. The changes in the manuscript have been highlighted in red.
Below we sequentially address all of the points raised by the Reviewers.
Reviewer 2:
Major Concerns:
Comment 1: Regarding the number of participants, the data on Abstract (200 participants), Methods (62 children in each subgroup) and Figure 2 (128 participants) is contradictory.
Response: We sincerely apologize for the oversight. The error has been corrected, and the data has been harmonized.
Comment 2: Why authors only present anthropometric data for the characterization of the sample? Information regarding age, sex and place of residence will be crucial for readers.
Response: Table 1 has been supplemented with the indicated data.
Comment 3: Lines 215-217: “A comparison of blood test results taken one year after the educational intervention showed that the experimental group had significantly higher levels of ferritin and lower levels of TIBC and ferroportin”. This information is not in accordance with results presented on Table 2.
Response: Thank you for your observation. We have provided clarification in the manuscript.
A comparison of blood test results taken one year after the educational intervention (follow up vs. follow up) showed that the study group had significantly higher levels of ferritin and lower levels of TIBC and ferroportin.
Comment 4: A single repeated measures analysis (RM-ANOVA) or a Mixed Models approach would be more appropriate, powerful, and informative than calculating multiple point-wise differences and conducting separate comparisons (Table 2 and 3). Also, since the study includes both pre- and post-test measurements, comparing dietary intake only at the end of the trial is misleading. Differences at baseline are not accounted for, and the analysis does not assess changes over time (Table 4).
Response: The use of delta values (i.e., change scores calculated as post-intervention minus baseline values) is an accepted method for assessing within-group changes and between-group differences in intervention studies (1,2). While repeated measures ANOVA (RM-ANOVA) accounts for time-dependent correlations and interaction effects, delta calculation offers a simpler approach that is widely used.
- Woźniak, D.; Podgórski, T.; Krzyżanowska-Jankowska, P.; DobrzyÅ„ska, M.; WichÅ‚acz-Trojanowska, N.; PrzysÅ‚awski, J.; DrzymaÅ‚a-Czyż, S. The Influence of Intensive Nutritional Education on the Iron Status in Infants. Nutrients 2022, 14, 2453, doi:10.3390/nu14122453.
- Jamka, M.; Czochralska-DuszyÅ„ska, A.; MÄ…dry, E.; Lisowska, A.; JoÅ„czyk-Potoczna, K.; Cielecka-Piontek, J.; BogdaÅ„ski, P.; Walkowiak, J. The Effect of Conjugated Linoleic Acid Supplementation on Densitometric Parameters in Overweight and Obese Women—A Randomised Controlled Trial. Medicina 2023, 59, 1690, doi:10.3390/medicina59091690.
Comment 5: It is crucial that authors present data regarding the Parental engagement, as it influences the results of the outcomes.
Parental activity was monitored using a dedicated mobile application, which tracked app usage frequency. Throughout the intervention, parents also regularly consulted with a certified dietitian to reinforce the educational message. Additionally, all participating parents completed and submitted a dietary questionnaire, confirming their commitment to the study.
Comment 6: In the tables, both mean and median values are presented together with a single p-value. This may be confusing, since the statistical test applied (and therefore the p-value) should correspond to the main measure of central tendency chosen, depending on the distribution of the data. It would be clearer to either: a) select one primary measure (mean or median, depending on data distribution) and present the p-value from the appropriate test or (b) present both descriptive measures but specify explicitly which one guided the statistical comparison."
Response: Both medians and means were presented in the manuscript according to CONSORT guidelines. We also provided missing information in the text.
….. All results were subjected to statistical analysis. Data are presented as medians with interquartile ranges (1st–3rd quartiles), as well as means with standard deviations (SD) and 95% confidence intervals (95% CI) in accordance with the CONSORT statement. Normality of data distribution was assessed using the Shapiro–Wilk test. In cases where normality was not confirmed, the Mann–Whitney U test or χ2 test was applied to compare results between groups. To evaluate intragroup changes between pre- and post-intervention values in both study and control groups, either the paired sample t-test or the Wilcoxon signed-rank test was used, depending on data distribution. For variables with a normal distribution, data were analyzed using means and standard deviations. For variables with a non-normal distribution, medians and interquartile ranges were used. Additionally, to evaluate the magnitude of intra-group changes in the analyzed biochemical parameters, delta values (Δ) were calculated as the difference between post- and pre-intervention measurements.
Comment 7: What are the main differences between this study and the already published study (reference 19)?
Response: Our previous study demonstrated that general parental nutritional education (not specifically focused on iron intake) had a positive impact on iron metabolism parameters in infants. The study was conducted among 200 parents whose children underwent a year-long nutritional intervention starting from the 6th week of life. Parental education resulted in improvements in iron-related metabolic indicators, including ferritin, transferrin, red blood cell count (RBC), hemoglobin (HGB), mean corpuscular volume (MCV), and hematocrit (HCT). However, likely due to the general nature of the educational content, 25% of infants in the intervention group still exhibited hemoglobin levels below the reference range. Therefore, in the present study, we implemented a targeted nutritional education program aimed at improving iron metabolism and included the analysis of more specific markers of iron deficiency.
Minor comments:
Comment 1: Line 21: Authors must change “and the control group without any intervention.” by “and the control group, which received basic infant nutrition guidelines
Response: Thank you for your suggestion. The change has been made.
Comment 2: Lines 19-23: Authors must clarify the methods used to analyse the intake of nutrients and biochemical parameters.
Response: Thank you for your comment.
Plasma concentrations of iron metabolism parameters—among others iron, ferritin, ferroportin, and total iron-binding capacity (TIBC)—were assessed at both the beginning and end of the study. Additionally, at the final time point, dietary intake of iron and components influencing its absorption (e.g., vitamin C, fiber, etc.) was evaluated based on food diaries completed by the parents.
Comment 3: Line 26: “TIBC value”. Authors must add the meaning of this abbreviation.
Response: Thank you for your observation. We have now clarified the meaning of the abbreviation “TIBC” (Total Iron Binding Capacity) in the manuscript.
Comment 4: On Abstract, authors must refer the direction of the results presented. The way i tis written do not allow readers to understand which group obtained the better results.
Response: The results section has been revised:
At the end of the study, the study group showed a significantly higher level of ferritin (p = 0.0067) and lower levels of TIBC (p = 0.0478) and ferroportin (p = 0.0410) compared to the control group. Moreover, infants in the study group demonstrated significantly higher intake of both iron (p = 0.0252) and vitamin C (p = 0.0458).
Comment 5: Lines 40-41: It will be important that authors presente results beyond the ages of 6 to 12 months.
Response: The following sentence has been added:
Among toddlers over 12 months of age living in developed countries (such as the United States, New Zealand, Finland, and Greece), the prevalence of iron deficiency without anemia may reach up to 30%) [13–16].Globally, the issue of iron deficiency and the resulting anemia is even more widespread. According to WHO data, the global prevalence of IDA among preschool-aged children (0–4.99 years) is 42.6% (95% CI: 37.7–47.4). The highest prevalence was recorded in Africa, at 62.3% (95% CI: 59.6–64.8). Among all African countries, anemia was most prevalent in Burkina Faso, where the rate reached 86% (95% CI: 82–89). High rates of anemia were also observed in South-East Asia, where the prevalence of IDA was 53.8% (95% CI: 39.9–63.9), including as much as 61% (95% CI: 51–69) in Pakistan [17,18].
Comment 6: Line 81 and Figure 1: “infants less than 8 weeks of age,” and “up to age of 12 weeks”. There is a discrepancy in this inclusion criteria. I kindly ask authors to correct it.
Response: data corrected.
Comment 7: Lines 96-97: Authors must insert the exact duration of the intervention. (in the case “12 months”)
Response: The following sentence has been added: The nutritional intervention lasted until the infant reached 12 months of age.
Comment 8: On Figure 1: authors must refer the exact duration of the intervention and not present na interval between 11 and 13 months.
Comment 9: On Figure 1, authors must also presente information regarding the control group (intervention and number of participants)
Comment 10: On outcomes, authors must insert the “intake of nutrientes”.
Response: Figure 1 has been corrected.
Comment 11: There is a discrepancy in the outcomes presented and the outcomes presented on biochemical measurements.
Response: We have provided clarification in the manuscript.
The biochemical assessment of iron status included the following parameters: serum iron concentration, ferritin, hepcidin, ferroportin, TIBC, UIBC, and hs-CRP.
Comment 12: Line 156. Not only at the end of the study but also in the beginning, correct?
Response: The assessment of nutrient intake was conducted only at the end of the study. This information has been updated in the Figure 1.
Comment 13: Lines 177-178: Accordingly, 62 children will be recruited and randomly assigned to each subgroup.” This is not a protocol, this is not a study that will be performed. In the presente study 200 participantes were included in the presente study. I kindly ask authord to check this sentence.
Response: Thank you for pointing this out. The sentence has been revised accordingly.
Comment 14: On Figure 2, authors must change “Excluded (N= 128)” by “Excluded (N=13)”.
Response: Figure 2 has been corrected
Comment 15: Lines 192-193. Authors refer to 8 parents that failed to attend the post-study. However on Figure 2, information stated 11 parents (6 from intervention and 5 from control group)
Response: Thank you for identifying our error. The data have been revised accordingly.
Comment 16: Authors must uniformize the terminology (experimental group OR study group) and (infants VS. young children). In the first case, authors can choose between the two valid options. In the second case, regarding the scope of the study, infants will be more correct.
Response: The terminology was uniformize (the names: study group and infants were adopted).
Comment 17: On the second paragraph of Discussion, authors must also refer the importance of dietary fibers and proteins.
Response: Thank you for your valuable suggestion. We have now expanded the second paragraph of the Discussion section to include the role of dietary fiber and protein in infant nutrition.
The average fiber intake in both groups was within the normal range. Adequate fiber consumption is extremely important. While normal fiber intake does not interfere with iron absorption [43], excessive fiber consumption—often associated with high dietary levels of phytic acid—can significantly reduce iron bioavailability. Phytates may decrease iron absorption by up to 50% [44]. On the other hand, the benefits of dietary fiber are increasingly emphasized. Early exposure to fiber-rich foods may support long-term gastrointestinal health and the development of healthy eating habits. Dietary fiber contributes to digestive health by promoting the growth of beneficial gut microbiota, which colonize the intestines most intensively during infancy. Acting as a prebiotic, fiber nourishes bacteria of the genus Lactobacillus (e.g., Lactobacillus plantarum 299v), which enhance nutrient availability for intestinal epithelial cells and other commensal bacteria [45].
Our study also assessed protein intake in the infants’ diet. The total protein intake was excessive in both groups—nearly three times higher than the recommended daily allowance. Although excessive protein intake is concerning, our findings are consistent with European studies showing that protein consumption among infants often exceeds recommended levels [15,26,46–49]. In our study, protein intake was primarily evaluated in the context of its role as a source of heme iron. While promoting meat consumption in infancy may contribute to protein overconsumption, studies assessing the impact of meat intake on infant growth have yielded mixed results [50].
Comment 18: In the beginning of Discussion, authors must presente the main results.
Response: The paragraph has been added:
Our study demonstrated that parental nutritional education positively influences iron metabolism in infants. Infants in the intervention group not only consumed higher amounts of iron and vitamin C, but also exhibited higher plasma ferritin concentrations and lower levels of iron deficiency markers (TIBC and ferroportin). Additionally, the mean increase in ferroportin concentration (expressed as delta value) was lower in the intervention group.
Round 2
Reviewer 1 Report
Comments and Suggestions for Authors
The revision has substantially improved the manuscript, with clearer background, methodology, and discussion. Although the trial registration is lacking and tables could be simplified, these do not detract from the overall contribution.
Author Response
Response to the comments made by the reviewers
Manuscript ID: nutrients-3836723
Title: The influence of intensive nutritional education on the iron status in infants - randomised controlled study
Dear Reviewer, we appreciate all your insightful comments. Thank you for your suggestions.
Below we sequentially address all of the points raised by the Reviewers.
Reviewer 1:
Comments: The revision has substantially improved the manuscript, with clearer background, methodology, and discussion. Although the trial registration is lacking and tables could be simplified, these do not detract from the overall contribution.
Response:
We sincerely apologize for not registering the study in advance. We acknowledge the importance of preregistration for ensuring transparency and scientific rigor, and we are committed to adhering to this practice in future projects. We do not want to do reverse registration.
We are afraid of simplifying the data presented in Tables because we want to show as much information as possible.
Reviewer 2 Report
Comments and Suggestions for Authors
Dear Authors:
Regarding the manuscript with title “ The influence of intensive nutritional education on the ironstatus in infants - randomised controlled study”, I still have major concerns that hinders its publication.
Comment 1: I thank the authors the response to my previous comment regarding parental adherence. The manuscript describes well the methods used to monitor parental adherence, such as tracking app usage frequency, regular consultations with a certified dietitian, and completion of dietary questionnaires. However, it is unclear what the actual levels of parental adherence were during the intervention Without this information, it is impossible to fully interpret the effectiveness of the intervention and to draw reliable conclusions about its impact on infant iron status."
Comment 2: The study group, which received intensive nutritional education around 12 months (Lines 19-20). and “ The nutritional intervention lasted until the infant reached 12 months of age.”(lines 107-108). There is an inconsistency between the two statements regarding the duration of the nutritional intervention. One sentence suggests that the study group received intensive nutritional education for around 12 months, while the other states that the intervention lasted until the infant reached 12 months of age.
Comment 3: The manuscript reports that nutritional information was collected only at the end of the study to compare the intervention and control groups. While this approach may offer a snapshot of differences at that time point, it significantly limits the ability to assess the true impact of the intervention. Without baseline data, it is impossible to confirm whether the groups were comparable at the start or to measure changes within groups over time.
Comment 4: I thank the authors for the clarification regarding the use of delta values. Indeed, calculating change scores is a common and accepted method for assessing intervention effects. However, it is worth noting that approaches such as repeated measures ANOVA or Mixed Models provide a more robust framework by explicitly modeling the correlation between repeated measures and allowing for testing of interaction effects between time and group. These methods can increase statistical power and give a clearer understanding of how dietary intake changes over time within and between groups. While the delta approach simplifies the analysis, it may overlook nuances such as time-dependent effects or variability in baseline measurements that RM-ANOVA or Mixed Models can capture.
Comment 5: Lines 275-277: A comparison of blood test results taken one year after the educational intervention (baseline vs. follow up) showed that the study group had significantly higher levels of ferritin and lower levels of TIBC and ferroportin. This is not in line with the results presented on Table 2
Author Response
Response to the comments made by the reviewers
Manuscript ID: nutrients-3836723
Title: The influence of intensive nutritional education on the iron status in infants - randomised controlled study
Dear Reviewer, we appreciate all your insightful comments. Thank you for your suggestions.
Below we sequentially address all of the points raised by the Reviewers.
Reviewer 2:
Comment 1: I thank the authors the response to my previous comment regarding parental adherence. The manuscript describes well the methods used to monitor parental adherence, such as tracking app usage frequency, regular consultations with a certified dietitian, and completion of dietary questionnaires. However, it is unclear what the actual levels of parental adherence were during the intervention Without this information, it is impossible to fully interpret the effectiveness of the intervention and to draw reliable conclusions about its impact on infant iron status."
Response: We appreciate the reviewer’s observation regarding the need for more detailed information on parental adherence. We added the following clarification:
The readability rate of nutritional messages among parents in the intervention group was high, averaging 88% ± 25%. Notably, 70% of parents read all the nutritional messages sent during the study period, while approximately 5% read up to only 10% of the messages.
Comment 2: The study group, which received intensive nutritional education around 12 months (Lines 19-20). and “ The nutritional intervention lasted until the infant reached 12 months of age.”(lines 107-108). There is an inconsistency between the two statements regarding the duration of the nutritional intervention. One sentence suggests that the study group received intensive nutritional education for around 12 months, while the other states that the intervention lasted until the infant reached 12 months of age.
Response: We standardized the information in the manuscript.
…The parents of 115 infants were randomly assigned to two groups: the study group, which received intensive nutritional education up to 12 months of age, and the control group, which received basic infant nutrition guidelines…
…Nutritional intervention lasted until 12 months of age…
Comment 3: The manuscript reports that nutritional information was collected only at the end of the study to compare the intervention and control groups. While this approach may offer a snapshot of differences at that time point, it significantly limits the ability to assess the true impact of the intervention. Without baseline data, it is impossible to confirm whether the groups were comparable at the start or to measure changes within groups over time.
Response: Our experience indicates that measuring the intake of individual macro- and micronutrients in infants during the first months of life is subject to considerable error. It is easier for parents to record nutritional information from jar labels and measure the amount of formula milk consumed from a bottle than to accurately estimate the intake of breast milk based on weighing the infant before and after feeding—especially since these feedings can be quite lengthy during the early months.
Nevertheless, we estimated protein and iron intake based on parental records for the enrolled infants. The percentage of daily recommended intake for these nutrients did not differ significantly between the study groups (e.g. SG vs CG: Fe – mean ± SD – 70.0 ± 69.5% vs 63.5 ± 47.9%, p = ns; protein – 164.2 ± 127.2% vs 225.0 ± 195.5%, p = ns).
We acknowledge that the lack of significant differences is primarily due to high statistical error; therefore, we chose not to present these data in the manuscript.
Comment 4: I thank the authors for the clarification regarding the use of delta values. Indeed, calculating change scores is a common and accepted method for assessing intervention effects. However, it is worth noting that approaches such as repeated measures ANOVA or Mixed Models provide a more robust framework by explicitly modeling the correlation between repeated measures and allowing for testing of interaction effects between time and group. These methods can increase statistical power and give a clearer understanding of how dietary intake changes over time within and between groups. While the delta approach simplifies the analysis, it may overlook nuances such as time-dependent effects or variability in baseline measurements that RM-ANOVA or Mixed Models can capture.
Response: We thank the reviewer for this valuable comment and fully acknowledge that repeated measures ANOVA and Mixed Models offer a more sophisticated framework for analyzing longitudinal data, particularly by accounting for within-subject correlations and interaction effects over time.
In our study, we chose to calculate delta values (i.e., the difference between baseline and endpoint measurements) as a pragmatic approach, primarily due to the simplicity of interpretation and the nature of our dataset. Given the limited number of time points and the relatively small sample size, we considered the delta method appropriate for capturing the overall change in dietary intake and iron status.
We agree that more advanced statistical models could provide additional insights, especially regarding time-dependent effects and baseline variability. This is an important consideration for future studies with larger cohorts and more frequent measurements. We have added a note in the discussion to acknowledge this limitation and to suggest the use of repeated measures models in future research.
Additionally, in future studies with larger cohorts and more frequent measurements, it is worth using more advanced statistical models (e.g. repeated measures ANOVA), which may provide additional information, especially in terms of baseline variability and time-dependent effects.
Comment 5: Lines 275-277: A comparison of blood test results taken one year after the educational intervention (baseline vs. follow up) showed that the study group had significantly higher levels of ferritin and lower levels of TIBC and ferroportin. This is not in line with the results presented on Table 2.
Response: Thank you for your observation. The results were compared follow up vs. follow up as shown in line 276. The data have been carefully reviewed and, according to the authors, are correct. Ferritin concentration measured at follow-up (study group vs. control group) was higher in the study group (median: 35.1 ng/mL vs. 18.9 ng/mL, p = 0.0067). In contrast, TIBC and ferroportin concentrations were lower in the study group (TIBC median: 356.5 µg/dL vs. 382.0 µg/dL, p = 0.0478; ferroportin median: 22.8 ng/mL vs. 32.8 ng/mL, p = 0.0410).
Round 3
Reviewer 2 Report
Comments and Suggestions for Authors
Dear Authors.
I still have two major concerns that hinder the publication of this manuscript.
Comment 1:
While I understand the practical challenges of accurately assessing nutrient intake in infants—particularly in those who are breastfed—the absence of any baseline nutritional data significantly limits the ability to assess the impact of the intervention. The justification provided does not fully address this key methodological concern.
Comment 2:
I would like to point out what I consider to be a significant methodological flaw in the study design. According to the inclusion criteria, infants were recruited at less than 12 weeks of age. However, the intervention was delivered until 12 months of age, meaning that the duration of exposure to the intervention can vary among participants (no data is available regarding the age of infants). For example, an infant enrolled at 4 weeks would receive approximately 11 months of intervention, while one enrolled at 11 weeks would only receive about 9 months. This variation in exposure time introduces a potential bias, as the effectiveness of the intervention may be influenced by the duration of participation. Furthermore, the statistical analysis section does not indicate any adjustment or control for this variation
Author Response
Manuscript ID: nutrients-3836723
Title: The influence of intensive nutritional education on the iron status in infants - randomised controlled study
Dear Reviewer, we appreciate all your insightful comments. Thank you for your suggestions.
Below we sequentially address all of the points raised by the Reviewers.
Comment 1: While I understand the practical challenges of accurately assessing nutrient intake in infants—particularly in those who are breastfed—the absence of any baseline nutritional data significantly limits the ability to assess the impact of the intervention. The justification provided does not fully address this key methodological concern.
Response: An additional clarification has been included in subsection 2.8. Dietary Intake:
Parents were also given the opportunity to prepare a preliminary dietary record at the time of the infant’s enrollment in the study. Based on our experience, the assessment of macro- and micronutrient intake in infants during the first months of life is subject to considerable error. It is generally easier for parents to record nutritional information from jar labels and measure the quantity of formula consumed from a bottle than to accurately estimate breast milk intake using pre- and post-feeding weight measurements—particularly given that breastfeeding sessions can be lengthy during early infancy (supplementary data).
Nevertheless, we estimated the intake of iron, protein, and vitamin C based on the data provided by parents. Due to the high statistical error associated with these estimates, the results are presented solely as supplementary data.
Comment 2: I would like to point out what I consider to be a significant methodological flaw in the study design. According to the inclusion criteria, infants were recruited at less than 12 weeks of age. However, the intervention was delivered until 12 months of age, meaning that the duration of exposure to the intervention can vary among participants (no data is available regarding the age of infants). For example, an infant enrolled at 4 weeks would receive approximately 11 months of intervention, while one enrolled at 11 weeks would only receive about 9 months. This variation in exposure time introduces a potential bias, as the effectiveness of the intervention may be influenced by the duration of participation. Furthermore, the statistical analysis section does not indicate any adjustment or control for this variation
Response: An additional explanation has been added to subsection 2.2. Participants:
To ensure maximum comfort during participant recruitment and to include only healthy infants (without chronic conditions or seasonal infections), recruitment was primarily conducted during the first vaccination visit. As a result, 95% of infants were enrolled at 6 weeks of age, or at 8 weeks in cases of delayed appointments. The maximum age at enrollment was 12 weeks.
We would also like to emphasize that both the timing of enrollment and the duration of participation were accounted for in the statistical analyses. In addition to birth weight, we present Z-scores for body weight at the time of enrollment. Furthermore, data from dietary records were recalculated as a percentage of daily nutritional requirements, individually determined for each participant.